# Tocilizumab demonstrates superiority in decreasing C-reactive protein levels in hospitalized COVID-19 patients, compared to standard care treatment alone

Carolina Calderón-Ochoa,[1] Narek Plamenov-Donchev,[1] Fernando Hernández-Quiñones,[1] Olga Mendoza-López,[1] Luis Carlos Hinojos-Gallardo,[1] Asunción José Longino-Gómez,[2] Raúl Hernádez-Saldaña,[2] Jorge Duque-Rodríguez,[1] María Cecilia Ishida-Gutiérrez[1]

**ABSTRACT** Severe acute respiratory syndrome coronavirus 2 has caused a global pandemic, leading to health, economic, and political crisis. The virus triggers the activation of inflammatory reactants including interleukin-6 (IL-6), ferritin, and C-reactive protein (CRP), causing multiorgan damage, particularly affecting the lungs. Tocilizumab, an IL-6 receptor blocker, has the potential to diminish the progression of the disease and reduce organ damage and long-term complications. The aim of this observational retrospective cohort study was to evaluate the efficacy of tocilizumab in decreasing CRP levels in hospitalized coronavirus disease 2019 (COVID-19) patients compared to standard care without the drug. The study included 141 patients during their Hospital Stay (HS), with 100 in the Tocilizumab group and 41 in the non-Tocilizumab group. Clinical information was collected from the electronic clinical record, analyzed using statistical software, and homogenized the CRP levels from the severe group to the levels of the less complicated group at 48 h of hospitalization. The results showed a statistically significant greater decrease in CRP levels in the Tocilizumab group at 48 h after the use of the treatment, with no differences in mortality or length of stay between the groups. In conclusion, tocilizumab accelerates the diminishing of CRP levels compared to standard treatment alone, and its use may have potential benefits in the management of severe COVID-19 patients when used alongside with follow-up quantification of CRP levels reduction.

**IMPORTANCE** Severe acute respiratory syndrome coronavirus 2 has caused a global pandemic, leading to health, economic, and political crises. International guidelines for managing coronavirus disease 2019 (COVID-19) give recommendations according to the severity of the disease and the level of oxygen therapy needed. Tocilizumab is an option for the therapeutic management of hospitalized patients with any level of oxygen therapy; IL-6 serum level is the parameter for the follow-up on the efficacy, but it is not available at many hospitals. In this study, we demonstrate that C-reactive protein determination can predict the response to tocilizumab in severe COVID-19, the target patients for treatment with this drug. The use of this affordable and extensively available biomarker supports clinical decisions for the early escalation of the therapy and for the rational use of this drug on those prone to improve with the use of it.

**KEYWORDS** COVID-19, C reactive protein, interleukin 6, tocilizumab

Address correspondence to María Cecilia Ishida-Gutiérrez, ishida.cecilia@gmail.com.

Carolina Calderón-Ochoa and Narek Plamenov-Donchev contributed equally to this article. Author order was determined alphabetically.

The authors declare no conflict of interest.

The severe acute respiratory syndrome coronavirus 2 (SARS-CoV-2) emerged in Wuhan, China, in December 2019 originating coronavirus disease 2019 (COVID-19). It rapidly spread worldwide and was classified as a pandemic state by the World Health

Organization, increasing the rate of hospitalizations and mortality and collapsing the healthcare system (1).

Up to 13 February 2023, 672,981,708 cases and 6,854,803 deaths have been reported worldwide due to COVID-19 (2). The risk factors for severe and critical COVID-19 infection are advanced age, male gender, comorbidities such as hypertension, diabetes type 1 and 2, obesity, chronic lung disease, and cancer (1). Additionally, the extension of lung damage seen on a computerized tomography (CT) scan is also related to higher levels of inflammatory markers such as C-reactive protein (CRP) and other severity indicators such as lymphopenia, neutrophilia, lower oxygen saturation and partial oxygen pressure (3).

The classification of patients into mild, moderate, severe, or critical stages is important to give priority to patients who may need intensive treatment. According to the World Health Organization, severe disease is defined as a transdermic oxygen saturation of less than 90% in ambient air, ventilatory frequency greater than 30 breaths per minute, and severe respiratory distress, while critical disease refers to the development of acute respiratory distress syndrome, septic shock, or other abnormalities that would require treatment with vasopressors (1). In other studies, the severe cases were defined as patients with <92% of oxygen saturation at room air and CRP ≥75 mg/L (4), or the admission to the intensive care unit (5).

The severity of COVID-19 has also been determined by serum biomarkers such as interleukin-6 (IL-6), CRP, ferritin, and lactate dehydrogenase due to their influence in the inflammatory process, leading to SARS-CoV-2 aggravation. These reactants are highly elevated during the severe stage of the disease with the following serum levels: IL-6 35–90 ng/mL, CRP 120–160 mg/L, ferritin 800–1,600 ng/mL, D-dimer 750–3,000 ng/mL, and lactate dehydrogenase 350–500 U/L; clinically, a ratio of arterial oxygen partial pressure to fractional inspired oxygen (PaO2/FiO2) between 100–200 mmHg is also associated with severe disease (6).

However, IL-6 effects may vary according to the stage of the disease. It plays a protective role during the asymptomatic, mild, and moderate stages, enhancing the antiviral response. But in the severe forms of disease, it decreases antiviral defenses by dysregulating the natural killer and cytotoxic CD8+ T cells; it may also prevent regulatory CD4+ T cells from differentiating and trigger a T helper 17 like response, leading to unrestrained hyperinflammation. This overreaction is also linked to lymphopenia, neutrophilia, vascular permeability, and possible complications like acute respiratory distress syndrome, coagulopathy, and multiorgan failure. Interestingly, in the critical stage of the disease, the IL-6 returns its protective role by leading a homeostatic response for tissue repair (6).

Tocilizumab is a monoclonal antibody of IL-6 membrane and soluble receptors used as treatment of some rheumatoid diseases and approved for the treatment of COVID-19 patients (4). Although it does not reduce serum IL-6 concentrations, it inhibits the patient's systemic inflammatory response when used during a severe stage, improving clinical outcomes; however, in nonsevere phases, it can also inhibit the anti-inflammatory effects of IL-6 and be counterproductive. The CRP levels could be used to evaluate the IL-6 bioactivity since this marker correlates better with the anti-inflammatory effects from IL-6 inhibitors in the severe stage of the disease; in fact, IL-6 levels increase after the blockage of IL-6 receptors as an expected result of their mechanism of action, but it could lead to misinterpretation of the inflammatory state (6).

CRP levels were used by various authors as criterion to administer tocilizumab; in a study, Issa et al. (7) used 150 mg/dL. In both studies, a significant diminish of the CRP levels was observed as a result of drug infusion, 5 and 3 days later, respectively. Furthermore, Khurshid et al. (8) concluded that a decrease of 50% or more in the CRP levels 48 h after tocilizumab application was an early and sensitive predictor of a good response to the drug, with the observation that 62.9% of their patients improved with the medication. In agreement with the anterior, Zacharias et al. (9) supported the use of CRP levels as survival predictor and noted that the reduction of this biomarker presented during the first 3 days after the application of the drug. Nonetheless, in most of these

studies, there was no control group, making it difficult to assess the effects of tocilizumab by itself. Additionally, the time lapse for the initial response to tocilizumab effects was not described.

The RECOVERY trial reported that tocilizumab administered in severe cases of COVID-19 improved survival at the 28 days follow-up, earlier discharge, and lower risk of progression to invasive ventilation (4); accordingly, REMAP-CAP found improved survival among intensive care unit (ICU) patients with tocilizumab treatment (5) and Sarhan et al. reported a lower mortality and lower progression to the ICU with the use of the drug. Still, another investigation conducted by Rosas et al. (10) did not find significant benefit in clinical status up to day 28 nor in mortality rate but found a possible reduction in time until discharge and duration at the ICU in the patients that received tocilizumab.

Therefore, it is important to understand whether the use of tocilizumab can improve the inflammatory response in severe cases of COVID-19 or not, and if these changes can be followed by quick and inexpensive tests such as CRP measurement. This could pose CRP quantification as a useful monitor of the inflammation reduction after the application of tocilizumab in severe patients to evaluate the improvement into mild or moderate severity stages. Consequently, in the present study, CRP levels were quantified, as well as other COVID-19 severity markers in hospitalized patients, in order to analyze the inflammatory response during the 48 h after receiving whether standard treatment alongside tocilizumab in severe patients, or standard treatment without tocilizumab in nonsevere patients.

## MATERIALS AND METHODS

### Study design and patients

This trial was conducted as an observational retrospective cohort single center study at Chihuahua, Mexico. The included patients were hospitalized due to COVID-19, confirmed by RT-PCR performed on nasopharyngeal swabs, or by positive serologic SARS-CoV-2 IgM and IgG antibodies. The involved patients in the present study were selected between 30 March 2020 and 02 November 2020 and were 18–85 years old at the time of admission. To avoid potential selection bias, all eligible patients who met the pre-specified inclusion criteria during the study period were included; pregnant, breastfeeding women and patients with cancer or active bacterial infections were excluded.

The sample was divided into two groups: the "Tocilizumab group" with patients who received standard care (antibiotics, anticoagulant, oxygen supply, and steroids if indicated) and the "Non-Tocilizumab group" which only received standard care without the drug. Since participants with or without Tocilizumab received a range of immuno-suppressive therapies that increase vulnerability to opportunistic bacterial infections, antibiotics were empirically initiated to provide broad coverage when there were signs of potential infection; this was done even if bacterial pneumonia was discarded by initial diagnostic testing but always keeping the balance between risk/benefit use of antibiotics on each patient.

The decision to administer tocilizumab or not was made according to physician's criteria based on blood CRP and ferritin levels along with transdermic oxygen saturation (SpO2), lung CT score, age, and comorbidities. In the imaging assessment, the national guidelines established by the "Instituto Nacional de Enfermedades Respiratorias" (INER) for CT severity score in COVID-19 classification were used to evaluate the severity and stage of the lung injury, according to the extent of the damage and the predominant pattern, respectively (11). Briefly, the severity of the injury is classified as mild (1–5 points), moderate (6–15 points), or severe (>15 points) depending on the extent of lesion by lung lobe assigning 1 point when the damage involves ≤5% of the lobe; 2 points > 5%–25% ; 3 points > 25%–50%; 4 points > 50%–75%; and 5 points if >75%. The stage of the lung injury is classified according to the predominant pattern in which ground glass opacity is linked with an initial stage of the disease, crazy paving with progression, and consolidation with the advanced stage.

The study was approved by the Research and Ethics Committee of "Hospital Angeles de Chihuahua" (18 CI-08 019 009) and (CONBIOETICA-08-CEI-001-20160413) respectively.

## Statistical analysis

After recollecting data in Epi-Info 7.2 and Excel, an exploratory analysis was performed on R studio. Subsequently, the relevant variables were processed on SPSS 28.0 and MINITAB 21 to complete the analysis. A $P$ value < 0.05 was recognized as statistically significant.

Statistical analyses specific to address the limitation of small sample size was employed, normality was assessed according to Kolmogorov-Smirnov and of severity, both in Tocilizumab and non-Tocilizumab groups (12). Spearman's correlation was applied to evaluate the association between the levels of CRP and CT score as well as with the SpO2 levels in both groups (13). Mann-Whitney's test was performed to analyze the behavior of the deltas from the CRP levels between groups (14).

A mixed design two-way, three levels ANOVA was conducted to evaluate a statistical difference of the CRP levels over time within groups (at admission, 24 and 48 h of hospital stay), CRP between treatment groups and CRP vs the interaction "time-treatment group" (15). The cut-off point used by Zizzo et al. (6) and Juarez et al. (11) to illustrate the therapeutic window in which patients are likely to benefit most from tocilizumab.

Afterward, a two-way, two levels ANOVA was implemented to identify the interval of time where there was a statistically significant difference of the CRP decrease between groups. Thus, it was performed the analysis of the CRP levels at admission vs 24 h, 24 vs 48 h, and admission vs 48 h of hospital stay according to the treatment group.

## RESULTS

### The Tocilizumab group and the non-Tocilizumab group were homogenous within their demographic characteristics

In this retrospective study, a total of 141 patients were included: 100 were treated with standard care plus tocilizumab and 41 received standard care without the drug. Standard treatment was composed of the following with the percentage of patients that received it specified in parenthesis: supplemental oxygen (100%), anticoagulant therapy (97.17%), steroids (96.4%), and antibiotics (65.95%). The median of hospital stay (HS) length was 7 days, with no significant differences between the groups ($P = 0.124$). There were 8 deaths (8%) in the Tocilizumab group, and 4 (9.75%) in the non-Tocilizumab group, without statistical significance in mortality between the groups ($P = 0.745$). Interestingly, this lack of difference in mortality was observed between the more severe Tocilizumab group and the non-severe non-Tocilizumab group after receiving the treatment as mentioned above.

The median age was 54 years and 66.66% of the patients were men. 84.3% were Hispanic and 15.6% Mennonites. Because of the nature of a single center study, the variability within the patients in terms of demographics and standard treatment was smaller in comparison to multi-center studies (10). The most frequent comorbidities were hypertension (34%), type 2 diabetes mellitus (27.6%), and obesity (58.86%). These main comorbidities were registered and the groups with or without Tocilizumab did not show significant difference (Table S1).

The body mass index (BMI) median was 31 kg/m$^2$, with a higher index in the Tocilizumab group ($P = 0.038$), but it was not linked to poor prognosis in either of the groups ($P = 0.271$ Tocilizumab group and $P = 0.972$ non-Tocilizumab) nor to increased HS length ($P = 0.196$ Tocilizumab and $P = 0.636$ non-Tocilizumab group, for further details refer to supplemental Material).

Up to 20% of the patients from the Tocilizumab group were admitted to the ICU in contrast to only 9.7% of the non-Tocilizumab group patients. ICU admission was

associated with death ($P < 0.001$) and greater HS length ($P = 0.010$) in the Tocilizumab group, but not in the non- Tocilizumab group ($P = 0.192$ and $P = 0.335$ respectively).

## The severity of the Tocilizumab group was higher than that of the on-Tocilizumab group

The severity of the groups at admission was significantly higher among the Tocilizumab patients according to their ferritin ($P < 0.001$) (Fig. 1A) with a median of 772 ng/mL in Tocilizumab and 274 ng/mL in non-Tocilizumab group; CRP levels ($P = 0.015$) (Fig. 1B) with a median of 219 mg/L in Tocilizumab group while in non-Tocilizumab group was 58.5 mg/L; and CT severity score ($P = 0.002$) (Fig. 1C) with a median in Tocilizumab group of 12 and 8 in non-Tocilizumab group; whereas SpO2 (Fig. 1D) at admission was not different between the groups ($P = 0.72$).

In the imaging criteria, the patients were subdivided based on CT severity score [mild (1–5) points, moderate (6–11, 16–19), and severe (12–15, 20–25)] and predominant

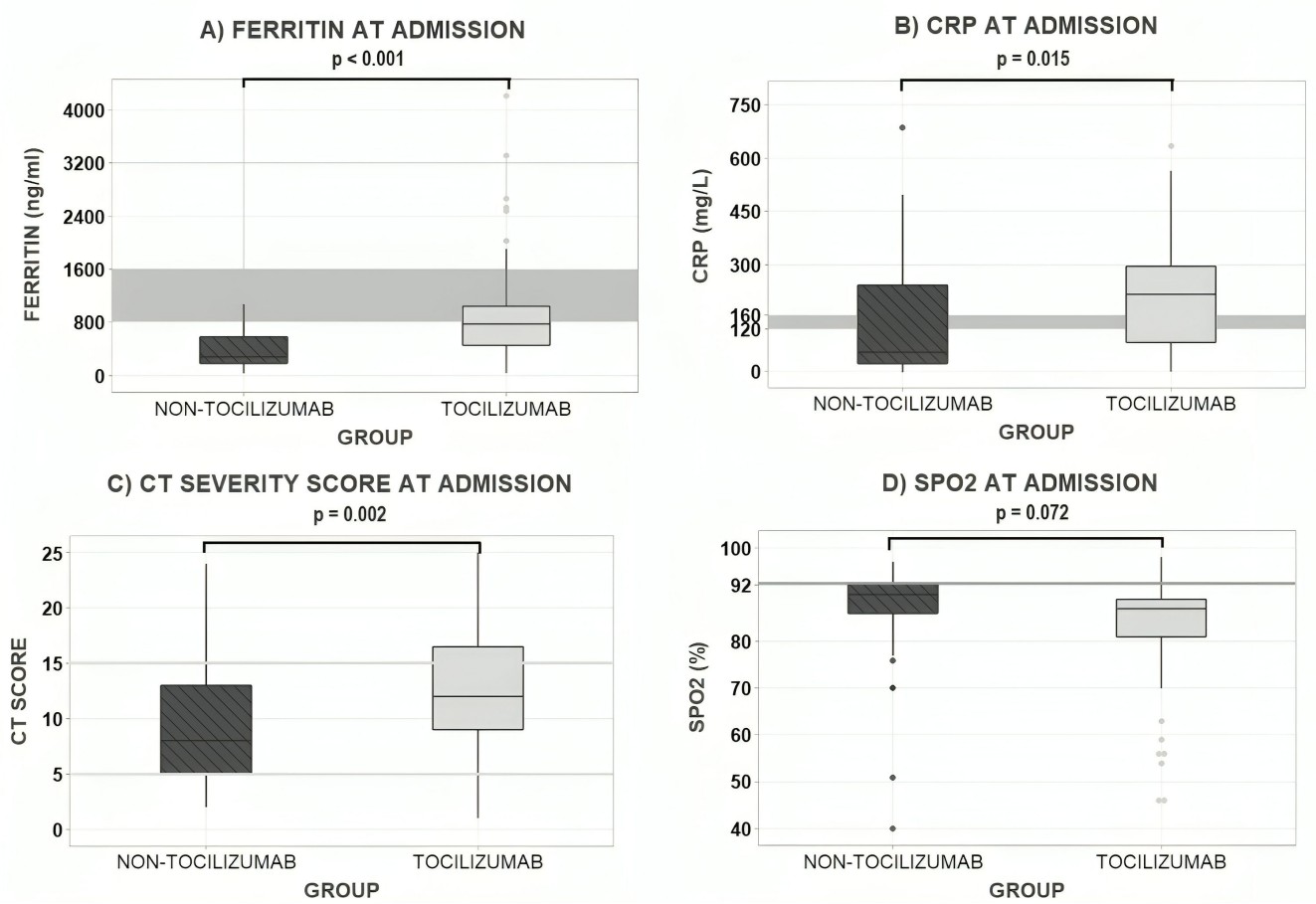

**FIG 1** Tocilizumab group was significantly more severe at admission compared to non-Tocilizumab group in terms of Ferritin, CRP, and CT score.(A) Box-plot of serum ferritin (ng/mL) divided by group, where the Tocilizumab group arrived with greater levels than non-Tocilizumab patients ($P < 0.001$). (B) Box-plot of blood CRP (mg/L) levels by group with higher levels presented in the Tocilizumab group ($P = 0.007$). (C) Box-plot of lung CT severity score vs group, with a significantly greater score observed among the Tocilizumab group based on the disease extension along pulmonary lobes ($P = 0.022$). (D) Boxplot of transdermic SpO2 (%) by group, where there is not a statistical difference between groups ($P = 0.072$). The gray horizontal lines represent the cutoff point that defines the severe stage of disease according to the literature in both laboratory (6) and imaging evaluation (11). Nonsevere patients received standard treatment, labeled here as non-Tocilizumab group, whereas severe patients, received tocilizumab alongside standard treatment, labeled here as Tocilizumab group. Comparisons between groups were analyzed with Student's *t* test. The indicators used to assess clinical severity due to COVID-19 infection were blood ferritin, blood CRP, lung computerized tomography (CT) score according to INER's classification and transdermic oxygen saturation (SpO2). Gray area represents the therapeutic window in which patients are likely to benefit the most from tocilizumab (Materials and Methods) for ferritin and CRP (A,B panels), cut off values for Ferritin and CRP and CT score are described on Materials and Methods.

pattern, following the guidelines established by the INER, where the severity score indicates the extent of lung injury and the predominant pattern indicates the progression stage of such injury; the ground glass opacity pattern indicates initial injury, crazy paving pattern a progressive damage, and the consolidation pattern an advanced lesion in the pulmonary tissue. In terms of the CT score, the Tocilizumab group had 29 (31.18%) patients classified as severe, 60 (64.51%) patients as moderated, and only 4 (4.3%) as mild cases; in contrast, the non-Tocilizumab group had 7 (20%) severe patients, 19 (54.28%) moderated, and 9 (25.71%) with mild severity (Fig. 2A). Severe patients tended to present crazy paving or consolidation as predominant patterns in both groups, whereas the mild cases commonly presented ground glass opacity as the main type of lesion. An

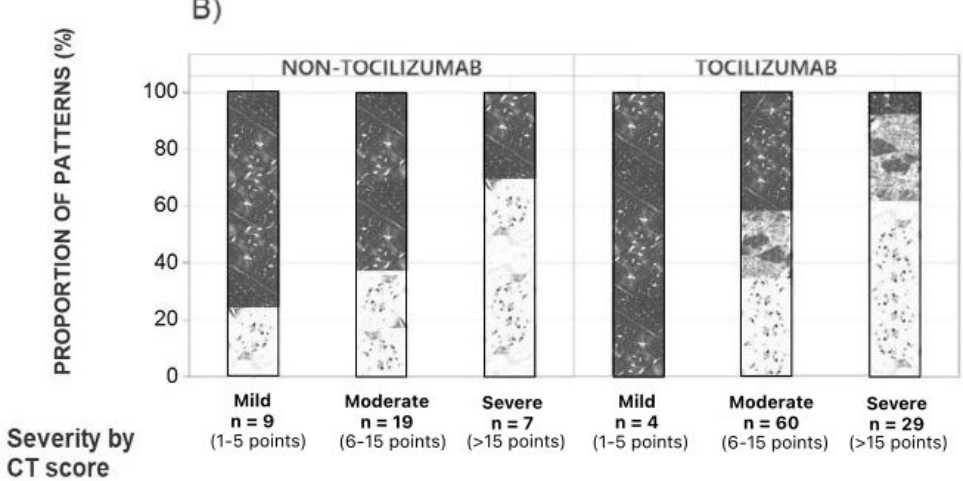

FIG 2 The Tocilizumab group includes patients with more severe CT scores and with a higher prevalence of advanced-stage lung injury associated with CT patterns, compared to the non-Tocilizumab group. (A) Data are presented as the number of patients divided by Tocilizumab and non-Tocilizumab group and subdivided by severity of the disease based on the lung CT score at admission. There was a significantly higher prevalence of moderate and severe patients among the Tocilizumab group according to Pearson's chi-squared ($P = 0.001$). (B) Data are presented as a proportion of the pattern according to the severity of their CT score, subdivided on Tocilizumab and non-Tocilizumab groups. A higher lung CT score was associated with crazy-paving and consolidation in both groups ($P < 0.001$), being these patterns the most common in the Tocilizumab patients, whereas the ground-glass opacity was the main pattern in the non-Tocilizumab group and was related with lower CT score.

elevated CT score was associated with death instead of discharge ($P = 0.002$ Tocilizumab and $P < 0.001$ non-Tocilizumab group) and also with lower SPO2 at admission ($P < 0.001$ in both groups). The prevalence of the CT patterns within Tocilizumab patients was 32.9% ground glass opacity, 25.5% crazy paving, and 41.4% consolidation; in the non-Tocilizumab group ground glass represented 60% of the patients and consolidation the remaining 40%, with no subjects that had predominance of crazy paving. The patterns of consolidation and crazy paving were more likely to predominate in the patients that were classified as moderate or severe according with their CT score in comparison to ground glass opacity that was the most frequent pattern among the mild patients in both groups ($P < 0.001$) (Fig. 2B); this indicates an association between the CT pattern and the severity of the disease.

According to the earlier data, the Tocilizumab group and the non-Tocilizumab group were homogenous within their demographic characteristics, but they presented a significant difference in terms of severity at their admission.

## Positive correlation between inflammatory markers and CT severity score

There was a significant correlation between the CRP levels and the CT severity score; a higher CRP correlates with greater CT score (Tocilizumab group $P = 0.002$ and non-Tocilizumab group $P = <0.001$), (Fig. 3A). This association was greater in the non-Tocilizumab group ($r = 0.660$) than the Tocilizumab group ($r = 0.333$). However, the CT pattern was not associated with higher CRP levels when analyzing the 123 patients that had a CT scan ($P = 0.076$).

The correlation between CRP at admission and SpO2 is meaningful in both groups ($P = 0.017$ Tocilizumab group and $P = 0.021$ non-Tocilizumab group) as shown in Fig. 3B where a greater CRP is correlated with lower SpO2 although there was no significant difference in the SpO2 between the groups ($P = 0.072$).

## After the application of Tocilizumab, the level of CRP decreased significantly, compared to the use of standard treatment in the non-Tocilizumab group

The CRP decrease, in the Tocilizumab group (230.97 mg/L at admission, down to 95.52 mg/L 48 h later), was more pronounced than in the non-Tocilizumab group (127.13 to mg/L at admission, down to 79.57 mg/L 48 h later). A mixed design two-way, three-level ANOVA was performed to analyze whether there was a significant difference in the decrease of CRP levels between groups besides its association with the course of time by itself based on the levels of CRP at admission, 24 and 48 h of hospital stay (HS).

A significant association was found between the decline in CRP levels over the course of time, without the influence of the groups ($F = 11.793$, $P < 0.001$), but there was not statistical significance difference in the CRP decline from the admission to the 48 h of HS between groups ($P = 0.166$, Fig. 4A). This can be explained by the fact that the medication's effect is not yet well integrated since its application was during the first 24 h of hospitalization.

Still, the slope between the CRP levels at admission vs 48 h later was remarkably steeper in the Tocilizumab group compared to the Non-Tocilizumab group. Therefore, a two-level analysis was performed to ratify the differences in slopes between groups according to the subdivision of time by intervals. In this mixed design ANOVA, it was found a significant difference in the decrease of the CRP levels from the 24 to 48 h of HS, between the Tocilizumab ($n = 23$) and non-Tocilizumab group ($n = 14$) ($F = 4.695$, $P = 0.129$), stating a statistically significant decrease in those patients who received tocilizumab. Neither the reduction of CRP at admission vs 24 h nor CRP at admission vs 48 h were significantly different among groups ($P = 0.930$ and $P = 0.037$, respectively; Fig. 4B).

To further analyze the changes in CRP levels and explore if there was a statistically significant change in the CRP decrease if the group received or not tocilizumab, the differences in CRP at 24 h minus CRP at admission ($\Delta$ 24–0 h), CRP at 48 h minus CRP at 24 h ($\Delta$ 48–24 h), and CRP at 48 h minus CRP at admission ($\Delta$ 48–0 h) were obtained

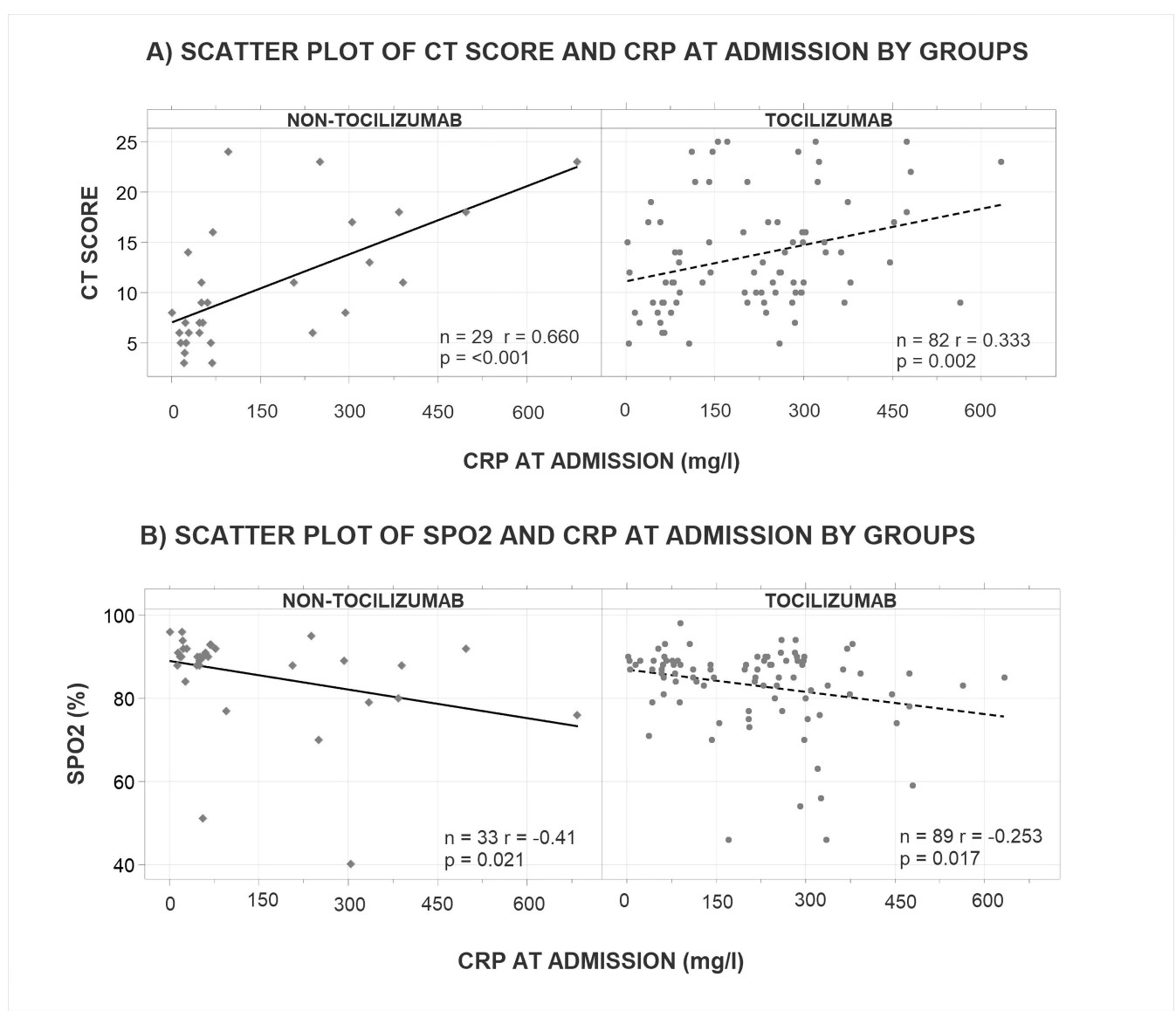

**FIG 3** Elevated CRP at admission is correlated with a higher CT score and SpO2, indicating advanced lung damage and poor oxygenation are associated with increased inflammatory response. (A) Scatter plot of CRP (mg/L) divided by group and according to their CT score at admission. In both groups, the levels of CRP are related to greater CT score, with a moderate correlation effect in the non-Tocilizumab patients and a mild effect among Tocilizumab patients. Due to higher CRP levels, there is a greater proportion of lung damage and higher CT scores in the Tocilizumab patients with a significant difference compared to the non-Tocilizumab group ($P = 0.001$). (B) Scatter plot of CRP (mg/L) divided by groups and according to their SpO2 (%). There is a significant association between CRP at admission and lower SpO2 (%) in both groups ($P = 0.017$ and $P = 0.021$ in Tocilizumab and non-Tocilizumab groups, respectively) although there is no significant difference in SpO2 between the groups ($P = 0.072$).

and then a Mann-Whitney *U* test was applied. A statistical significance was found in Δ 48–24 h between groups ($P = 0.011$) with greater decrease in the levels of the CRP in the Tocilizumab group, reflected in steeper slopes. The difference between groups in their Δ 24–0 h and Δ 48–0 h was not significant ($P = 0.876$ and $P = 0.110$, respectively; Fig. S1).

## DISCUSSION

The reduction in CRP levels was able to be compared between groups due to the use of only standard treatment in non-severe patients, labeled in the present report as non-Tocilizumab group; severe patients that received standard treatment alongside with tocilizumab in severe patients, labeled here as Tocilizumab group.

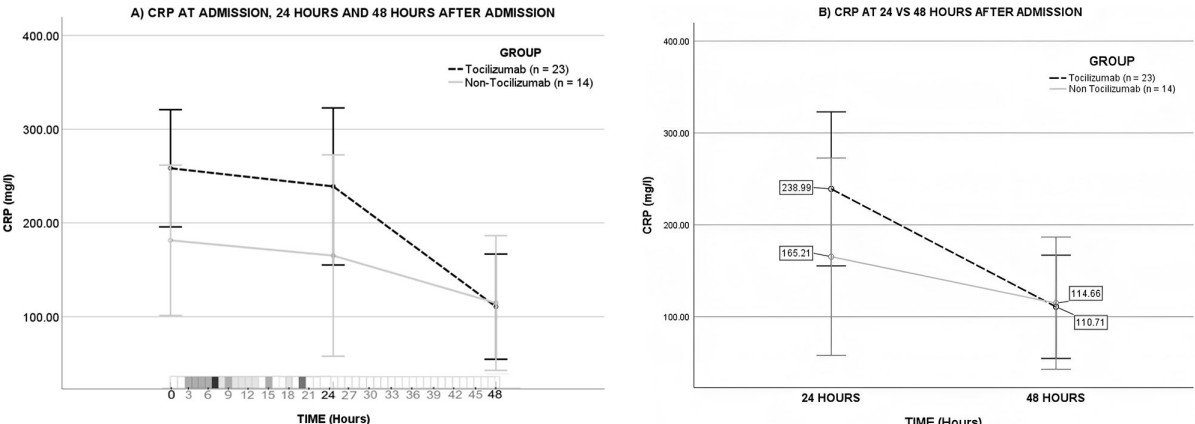

**FIG 4** Despite the Tocilizumab group presenting a more severe condition at admission, the treatment homogenized the CRP levels from the severe Tocilizumab group to the levels of the less complicated non-Tocilizumab group at 48 h of hospitalization. (A) Blood CRP (mg/L) means at admission, 24 h, and 48 h of hospital stay in the Non-Tocilizumab group ($n = 14$) and in the Tocilizumab group ($n = 23$). A two-way, three levels mixed design ANOVA was performed without statistically significant difference between groups ($P = 0.311$) nor within groups by time interaction ($P = 0.166$), but with a significant association in the decrease of the marker by time ($P < 0.001$). (B) The slope of the CRP levels from 24 h to 48 h of hospital stay between groups presented a significantly steeper decline ($P = 0.011$) within the Tocilizumab group in comparison to non-Tocilizumab patients after a two-way, two levels mixed design ANOVA was performed. The density of patients who received the first dose of tocilizumab in a specific time of hospital stay is presented in the scale bar that involves the first 48 h of HS. None of the patients included in this analisis received the first dose of tocilizumab after 21 h of HS nor more than one dose of the drug within the first 48 h after admission.

The unprecedented pandemic imposed logistic and physical barriers to conducting post-discharge assessments of hospitalized cohorts; therefore, the trial was designed with two critical endpoints: hospital length of stay and mortality rate. Without post-discharge patient follow-up, the durability of the reductions in inflammatory marker CRP cannot be confirmed. However, we emphasize that initial evidence of Tocilizumab among hospitalized COVID-19 patients demonstrating shortened hospital stays and lower mortality rates remains clinically impactful and that the significant drop in CRP levels allows to assess in a practical way the response to treatment with Tocilizumab on a specific level of severity of the disease group of patients.

The Tocilizumab group initially showed a higher probability of having a bad outcome, due to the higher levels of CRP and ferritin at admission, leading to a more extensive and advanced lung injury evidenced with a higher CT severity score and the predominance of crazy paving and consolidation CT patterns although both groups had similar SpO2 levels at admission (Fig. 1 and 2B). In line with this more severe presentation of COVID-19 in the Tocilizumab group, other laboratory markers such as lymphopenia and neutropenia, that are also linked to greater severity, were more pronounced (Table S1).

Despite the Tocilizumab group being more severe, there was no significant difference between groups in terms of mortality nor in HS length when compared to the non-Tocilizumab group (Table S2). This could be explained by the marked decrease in CRP levels of the Tocilizumab group as result of the drug's effect at the right stage of the disease, allowing the recovery from the lung injury and other organs affection after controlling the inflammatory state and, consequently, the homogenization of the more severe patients from the Tocilizumab group to behave as the less severe patients in the Non-Tocilizumab group. It has been pointed out that a decrease by 50% or more in CRP levels during the initial 48 h after the administration of tocilizumab is an indicator of a good response to the treatment with favorable survival rate (9). In this study, a mean reduction was observed of 53.67% of CRP, after only 24 h of the administration of the treatment in the Tocilizumab group patients ($n = 23$) that were analyzed with the mixed-design ANOVA in comparison with only 30.58% in the non-Tocilizumab patients ($n = 14$; Fig. 4B).

However, it is important to know that a beneficial decrease of the inflammatory state presents when the severity of the disease is at a point where the patient is classified as severe (CRP levels of 120–160 mg/L), excluding the mild, moderate, and even critical patients from the use of tocilizumab due to the counterproductive effects of the drug during these specific stages of COVID-19 where the IL-6 has an anti-inflammatory role (6). This is reinforced by the benefit of the use of tocilizumab within the patients that had a CRP >150 mg/L reported in CORINMUNO-TOCI- 1 (21). Due to the CRP initial levels, its decrease in the Tocilizumab group (230.97 mg/L at admission, down to 95.52 mg/L 48 h later) was more marked and favorable than the standard treatment by itself in the non-Tocilizumab group (127.13 mg/L at admission, down to 79.57 mg/L 48 h later). The importance of controlling the CRP levels relies on the fact that for every unit it increases, the risk of severe adverse events rises by 5% (22). The patients who were assigned to receive Tocilizumab had higher severity of the disease based on Ferritin, CRP, and CT scan indicators; however, the correlation between the CRP levels and the CT severity score was significative to both groups Tocilizumab group $P = 0.002$ and non-Tocilizumab group $P = <0.001$ (Fig. 3A); this association was more significant in the non-Tocilizumab group ($r = 0.660$) than on the Tocilizumab group ($r = 0.333$). These results eliminate a potential artificial correlation showing that sicker patients recruited and received Tocilizumab had a better correlation with the prognostic levels of CRP on the better response to treatment.

A limitation of this study resided in that it was only possible to account for the participation of one private center´s patients, and public hospitals had limited Tocilizumab prescriptions during the study timeframe due to its high cost; the included private hospital serves a broad population, and the ethnic groups present in this region were included (Table S1). Although the hospital is private, it receives patients from insurance companies. Therefore, there were no economic barriers to admitting patients of diverse socioeconomic groups. We also analyzed patient demographics and clinical characteristics to describe a profile that allows comparability to populations defined in other studies or potential beneficiaries of the conclusions derived from this study to build evidence supporting the conclusions and set the stage for more extensive multi-center trials.

On this retrospective study, the variability in the time lapse of the application of tocilizumab after the patient's admission to the hospital represented a restriction. A gray scale was decoded according to the number of patients that received a tocilizumab application by time among the Tocilizumab group, with darker colors indicating the most frequent time of drug administration after admission (Fig. 4). Note that there is some variation in the time of the medication's infusion, with 3 spikes of administration of tocilizumab at around 7, 15, and 20 h after hospital admission, yet the totality of the Tocilizumab group received the drug within the first 21 h of HS. This also explains why there was not a significant difference between groups in the decrease of the CRP levels from admission to 24 h of HS nor from admission to 48 h afterwards; this can be attributed to the latency in the onset of tocilizumab's effect after its administration.

When analyzing the stage and pattern in CT scans, it was found that greater CRP levels correlate with higher lung CT severity scores, as Tan et al. (23) described in their investigation. The Tocilizumab group had higher scores, with most of the patients classified as moderate or severe, while in the non-Tocilizumab group, the majority were moderate or mild patients. This higher imaging score is, in turn, related to a greater incidence of consolidation and crazy-paving patterns as predominant patterns in the lung CT. The Tocilizumab group showed a greater prevalence of consolidation, as a reflection of higher severity in the CT score, while the non-Tocilizumab group had the ground glass opacity as the most predominant pattern (see Fig. 2), which is linked with a lower CT severity score and, according to Mahdavi et al, it is also not associated with a worse outcome (24). Thus, the pattern observed on CT and its score can potentially provide information about the most beneficial time to block IL-6 inflammatory effects in cases where there is a predominance of consolidation or crazy-paving pattern due to their association with higher CT scores and its relationship with elevated CRP levels.

Microbiology Spectrum

Therefore, to achieve an integral treatment approach for patients, the assessment of laboratory parameters along with imaging features is the optimal strategy to identify the severe cases of

inflammatory effects of the IL-6 during this stage of the disease that cause progression of the damage, mainly lung injury, and even multiorgan damage (6).

Furthermore, the use of tocilizumab could prevent, or at least mitigate, future complications such as multiorgan dysfunction and fibrosis, with potential improvement of the outcomes seen in long-COVID phase. This potential benefit is important because according to Lopez et al. (25), up to 80% of the patients presented at least one symptom, within 14–110 days after viral infection: 34% of them had abnormalities in chest X-ray or in the CT, 8% had elevated CRP levels, 8% ferritin, and 3% IL-6.

The heterogeneous results in other studies (4, 10, 21) regarding tocilizumab's benefit in hospitalized patients due to COVID-19 could be the result of patient selection, sample sizes, the differences within the standard treatments, and tocilizumab administration in different stages of the COVID-19 disease. However, when tocilizumab is applied, the use of CRP as an indicator of its activity is especially useful due to its accessibility, low cost, and its close relationship to the modulation of the inflammatory status in hospitalized patients due COVID-19.

Other limitations of this study were that, due to the retrospective nature of this study, together with the onset of a new disease and the dynamic changes in therapeutic recommendations, it was difficult to homogenize the data between both groups. Yet the first application of tocilizumab was performed always in the first 20 h of HS. Also, because of the lack of serial laboratory measurements, the number of patients included in the Mixed-design ANOVA was reduced, and other biomarkers such as ferritin and D-dimer were not further evaluated.

Subsequent randomized case-control studies are recommended to evaluate CRP behavior during more than 48 h, in patients who have received tocilizumab as treatment, along with its impact on long-term mortality and hospital stay.

## Conclusion

The study results suggest that Tocilizumab is effective in reducing CRP levels in severe COVID-19 group to the levels of the less complicated group at 48 h of hospitalization. The results showed a statistically significant decrease in CRP levels in the Tocilizumab group, which indicates that the use of Tocilizumab accelerates the reduction of CRP levels. This is important because elevated levels of CRP are associated with a higher risk of complications, progression of disease, and worse prognosis in COVID-19 patients.

The study also found that there were no differences in mortality or length of stay between the Tocilizumab and the standard treatment groups. This suggests that Tocilizumab may not have a negative impact on the overall outcome of COVID-19 patients, and it may have potential benefits in the management of severe cases. However, further studies are needed to determine the long-term effects of Tocilizumab on COVID-19 patients, as well as its impact on mortality and other outcomes.

In conclusion, Tocilizumab accelerates the diminishing of CRP levels compared to standard treatment alone, and its use may have potential benefits in the management of severe COVID-19 patients when used alongside with follow-up quantification of CRP levels reduction.

## ACKNOWLEDGMENTS

This research is dedicated to the memory of our dear mentor and colleague, M.D. Jorge Duque-Rodriguez, who passed away while this paper was being written, proposed the original idea and protocol and was a pioneer in clinical research in COVID-19 since his perennial effort to help others with his knowledge unconditionally.

This research received no external funding.

C.-O.: Methodology, Software, Formal Analysis, Writing Draft; P.-D.: Metholdology, Visualization, Resources, Writing & Editing. M.-L.: Methodology, Formal Analysis, investigation, Data Curation, Writing Original Draft. H.-Q.: Software, Formal Analysis, Investigation, Data Curation Writing—Original Draft. L.-G., H.-G., H.-S.: Funding acquisition, Resources, Patients recruitment. D.-R.: Conceptualization, Methodology. I.-G.: Supervision, Conceptualization, Visualization, Writing—Review & Editing.

## AUTHOR AFFILIATIONS

[1]Autonomous University of Chihuahua, Faculty of Medicine and Biomedical Sciences, Laboratory of Pharmacoepidemiology, Chihuahua, Mexico
[2]Hospital Angeles Chihuahua, Critical Care Department, Chihuahua, Mexico

## AUTHOR ORCIDs

Carolina Calderón-Ochoa http://orcid.org/0000-0001-6458-1113
Fernando Hernández-Quiñones http://orcid.org/0000-0002-4120-7273
Olga Mendoza-López http://orcid.org/0000-0001-9067-1825
María Cecilia Ishida-Gutiérrez http://orcid.org/0000-0002-7642-2328

## ADDITIONAL FILES

The following material is available online.

### Supplemental Material

**Supplemental material (Spectrum02498-23-s0001.pdf).** Fig. S1; Table S1 and S2.

### Open Peer Review

**PEER REVIEW HISTORY (review-history.pdf).** An accounting of the reviewer comments and feedback.

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
