## [Reviewer comments · Microbiology Spectrum]

Microbiology Spectrum

Tocilizumab Demonstrates Superiority in Decreasing C-Reactive Protein Levels in Hospitalized COVID-19 Patients, compared to Standard Care Treatment Alone,

Carolina Calderón Ochoa, Narek Plamenov Donchev, Fernando Hernández-Quiñones, Olga Mendoza López, Luis Hinojos-Gallardo, Asunción Longino Gómez, Raúl Hernandez Saldaña, Jorge Duque Rodríguez, and María Ishida-Gutierrez

Corresponding Author(s): María Ishida-Gutierrez, Universidad Autonoma de Chihuahua Facultad de Medicina

Review Timeline:

Submission Date:	July 21, 2023
Editorial Decision:	October 18, 2023
Revision Received:	January 4, 2024
Editorial Decision:	January 27, 2024
Revision Received:	March 19, 2024
Accepted:	March 21, 2024

Editor: Maria Grazia Cusi

Reviewer(s): Disclosure of reviewer identity is with reference to reviewer comments included in decision letter(s). The following individuals involved in review of your submission have agreed to reveal their identity: Kamoru A ADEDOKUN (Reviewer #1); Kaan Çeylan (Reviewer #2)

Transaction Report:

DOI: <https://doi.org/10.1128/spectrum.02498-23>

October 18, 2023

Dr. María Cecilia Ishida-Gutierrez
Universidad Autonoma de Chihuahua Facultad de Medicina
Pharmacoepidemiology
Circuito Universitario 31109
Chihuahua, Chihuahua 31125
Mexico

Re: Spectrum02498-23 (Tocilizumab Demonstrates Superiority in Decreasing C-Reactive Protein Levels in Hospitalized COVID-19 Patients, compared to Standard Care Treatment Alone,)

Dear Dr. María Cecilia Ishida-Gutierrez:

Link Not Available

Sincerely,

Maria Grazia Cusi

Journals Department
Reviewer comments:

Reviewer #1 (Comments for the Author):

I want to thank the authors for their efforts. There are some unknown regarding this study most especially during post-treatment. Many areas demanding clarifications as follow.

While the study demonstrating the superiority of Tocilizumab in reducing C-Reactive Protein (CRP) levels in hospitalized COVID-19 patients compared to standard care treatment alone, may provide valuable insights, it's important to consider its potential weaknesses.

Small Sample Size: The study has a small sample size, the results may not be generalizable to a larger population. Small samples can lead to imprecise estimates and limited statistical power.

Lack of Adequate Randomization: I think because the number of patients was small, it therefore suffered the study to be more robust as the patients were not randomly assigned to various treatment groups based on their severity expected to be stratified by their proinflammatory markers and other anthropometric info, thus suggesting that selection bias could influence the presented results, as more sicker patients might be recruited and received Tocilizumab, which could artificially improve outcomes significantly.

Single-Center Study: The study was conducted at a single hospital or healthcare facility, the results may not be generalizable to other settings or populations. Apart from expecting enough sample size, one would also expect a multi-center study to generalize this outcome.

Short Duration: The short study duration might not capture the long-term effects or outcomes related to Tocilizumab treatment. It's important to consider the potential for delayed responses or long-term side effects.

Absence of Follow-up: Without follow-up, it is impossible to assess the sustainability of the observed effects. COVID-19 is a dynamic disease, and patient conditions may change over time. The investigators failed to inform us how the patients fared following the treatment. No follow-up information and so, the outcome could be a transient effect and worse still, no further attempts to substantiate other factors that might content the suggested mechanism.

Confounding Variables: The study should account for potential confounding factors that could influence CRP levels or clinical outcomes, such as comorbidities or other treatments administered alongside Tocilizumab.

Conflict of Interest: The authors also did not inform us whether there are any conflicts of interest, such as financial ties to the manufacturer of Tocilizumab, which could potentially bias the results or their interpretation.

Finally, I would suggest the authors remove any link connected to their private doc hub/archive such as google drive. All the figures should be provided along with the doc or as a supplementary.

Reviewer #2 (Comments for the Author):

Dear author,

The review of the study titled Tocilizumab Demonstrates Superiority in Decreasing C-Reactive Protein Levels in Hospitalized COVID-19 Patients, compared to Standard Care Treatment Alone has been completed. Your study has some shortcomings due to its retrospective nature, but the part examined has successful results. I have only one suggestion regarding your work.

It is stated in the materials and methods section that patients with bacterial pneumonia were not included in the study. However, in the findings section, it was stated that 65% of the patients received antibiotic treatment. It should be briefly stated why these patients received antibiotics. It will be sufficient to clarify this point.

I wish good work.

Staff Comments:

Preparing Revision Guidelines

Please return the manuscript within 60 days; if you cannot complete the modification within this time period, please contact me. If you do not wish to modify the manuscript and prefer to submit it to another journal, please notify me of your decision immediately so that the manuscript may be formally withdrawn from consideration by Microbiology Spectrum.

I want to thank the authors for their efforts. There are some unknown regarding this study most especially during post-treatment. Many areas demanding clarifications as follow.

While the study demonstrating the superiority of Tocilizumab in reducing C-Reactive Protein (CRP) levels in hospitalized COVID-19 patients compared to standard care treatment alone, may provide valuable insights, it's important to consider its potential weaknesses.

Small Sample Size: The study has a small sample size, the results may not be generalizable to a larger population. Small samples can lead to imprecise estimates and limited statistical power.

Lack of Adequate Randomization: I think because the number of patients is small, it therefore suffered the study to be more robust as the patients were not randomly assigned to various treatment groups based on their severity expected to be stratified by the proinflammatory markers, thus suggesting that selection bias could influence the presented results, as more sicker patients might be recruited and received Tocilizumab, which could artificially improve outcomes significantly.

Single-Center Study: The study is conducted at a single hospital or healthcare facility, the results may not be generalizable to other settings or populations. Apart from expecting enough sample size, one would also expect a multi-center study to generalize this outcome.

Short Duration: The short study duration might not capture the long-term effects or outcomes related to Tocilizumab treatment. It's important to consider the potential for delayed responses or long-term side effects.

Absence of Follow-up: Without follow-up, it is impossible to assess the sustainability of the observed effects. COVID-19 is a dynamic disease, and patient conditions may change over time. The investigators failed to inform us how the patients fared following the treatment. No follow-up information and so, the outcome could be a transient effect and worse still, no further attempts to substantiate other factors that might content the suggested mechanism.

Confounding Variables: The study should account for potential confounding factors that could influence CRP levels or clinical outcomes, such as comorbidities or other treatments administered alongside Tocilizumab.

Conflict of Interest: The authors also did not inform us whether there are any conflicts of interest, such as financial ties to the manufacturer of Tocilizumab, which could potentially bias the results or their interpretation.

Finally, I would suggest the authors remove any link connected to their private doc hub/archive such as google drive. All the figures should be provided along with the submitted doc or as a supplementary.

Dear Reviewers:

Thanks for the comments about the manuscript “Tocilizumab Demonstrates Superiority in Decreasing C-Reactive Protein Levels in Hospitalized COVID-19 Patients Compared to Standard Care Treatment Alone.” We found them very pertinent and enriched our manuscript. The responses to the kind suggestions and doubts are detailed below. We highlighted the sentences added or changed in the manuscript to track the changes quickly.

Respectfully,

Cecilia Ishida

Responses for reviewer 1 comments:

1. Small Sample Size: The study has a small sample size, the results may not be generalizable to a larger population. Small samples can lead to imprecise estimates and limited statistical power.

We appreciate the reviewer's comment regarding the limited sample size. Although we agree that the small sample size may impact the generalizability of the findings, **to avoid potential selection bias all eligible patients who met the pre-specified inclusion criteria during the study period were included and statistical analyses to address the limitation of small sample size was employed** (Section, Statistical Analysis) at adequate power that allow the data to be transferable to other clinical settings with similarity to the one described here to provide insight into the relation of low-cost proinflammatory markers with the ongoing clinical presentation and Tocilizumab response on SARS-CoV2. While a larger sample would have been ideal, the patients included were a consecutive series meeting the inclusion/exclusion criteria during a specified timeframe. We welcome suggestions from the reviewer regarding additional analyses to validate this study's findings further.

2. Lack of Adequate Randomization: I think because the number of patients was small, it therefore suffered the study to be more robust as the patients were not randomly assigned to various treatment groups based on their severity expected to be stratified by their proinflammatory markers and other anthropometric info, thus suggesting that selection bias could influence the presented results, as more sicker patients might be recruited and received Tocilizumab, which could artificially improve outcomes significantly.

The paper proposes using CRP as a marker to follow the efficacy of the Tocilizumab treatment; it is essential to recall that markers like IL-6 are not easily afforded by the budget of hospitals and patients in many countries of Latin America, like Mexico. In these countries or places with limited funds, we propose that the levels of CRP could be used as a biomarker, in the absence of more specific and expensive markers, to identify the patients who could receive benefit from the use of Tocilizumab and also as a predictor of a positive response to the drug during the following of the patient, guiding the decision of continuing

with the prescription or scale it to the next level indicated by international recommendations. Indeed, one section of the paper states that the patients who were assigned to receive Tocilizumab had higher severity of the disease based on Ferritin, CRP and CT scan indicators; however, the correlation between the CRP levels and the CT severity score was significant to both groups Tocilizumab group $p = 0.002$ and Non-Tocilizumab group $p = <0.001$ (Figure 3A), this association was more significant in the Non-Tocilizumab group ($r = 0.660$) than on the Tocilizumab group ($r = 0.333$). These results eliminate a potential artificial correlation showing that sicker patients recruited and received Tocilizumab had a better correlation with the prognostic levels of CRP on the better response to treatment.

3. Single-Center Study: The study was conducted at a single hospital or healthcare facility, the results may not be generalizable to other settings or populations. Apart from expecting enough sample size, one would also expect a multi-center study to generalize this outcome.

We appreciate the reviewer raising the point regarding the single-center nature of this study, certainly, a limitation of this study resides in that it was only possible to account for the participation of one private center's patients, and public hospitals had limited Tocilizumab prescriptions during the study timeframe due to its high cost; the included private hospital serves a broad population, the ethnic groups present in this region were included (Table S1). Although the hospital is private, it receives patients from insurance companies. Therefore, there were no economic barriers to admitting patients of diverse socioeconomic groups. We also analyzed patient demographics and clinical characteristics to describe a profile that allows comparability to populations defined in other studies or potential beneficiaries of the conclusions derived from this study to build evidence supporting the conclusions and set the stage for more extensive multi-center trials. A multicenter approach was preferred, as conducting such a study faces barriers at private hospitals, which infrequently collaborate on clinical trials on our country background. We agree that multi-center studies allow for increased generalizability across diverse settings and populations; since then, we have aimed to maximize this generalizability within the constraints of a single-center design as explained above.

4. Short Duration: The short study duration might not capture the long-term effects or outcomes related to Tocilizumab treatment. It's important to consider the potential for delayed responses or long-term side effects.

On great agreement with the comment, we do not intend to discuss on this paper the delayed responses or long-term side effects of the Tocilizumab, the study found that there were no differences in mortality or length of stay between the Tocilizumab and the standard treatment groups and we that Tocilizumab may not have a negative impact on the overall outcome of COVID-19 patients, and it had potential benefits in the management of severe cases. Intermediary outcomes were not able to be measured between groups. Further studies are needed to determine the long-term effects of off-label Tocilizumab use during

COVID-19 emergency status, although long-term effects of its use for other diseases are known.

5. Absence of Follow-up: Without follow-up, it is impossible to assess the sustainability of the observed effects. COVID-19 is a dynamic disease, and patient conditions may change over time. The investigators failed to inform us how the patients fared following the treatment. No follow-up information and so, the outcome could be a transient effect and worse still, no further attempts to substantiate other factors that might content the suggested mechanism.

We appreciate the reviewer raising the lack of patient follow-up as a limitation; as the reviewer mentioned, the unprecedented pandemic imposed logistic and physical barriers to conducting post-discharge assessments of hospitalized cohorts, therefore the trial was designed with two critical endpoints: hospital length of stay and mortality rate. Without post-discharge patient follow-up, the durability of the reductions in inflammatory marker CRP cannot be confirmed. However, we emphasize that initial evidence of Tocilizumab among hospitalized COVID-19 patients demonstrating shortened hospital stays and lower mortality rates remains clinically impactful and that the significant drop in CRP levels allows to assess in a practical way the response to treatment with Tocilizumab on a specific level of severity of the disease group of patients.

The lack of other longer-term outcome data remains an opportunity for further research, Tocilizumab availability was heavily constrained locally due to high costs and supply limitations; identifying hospitalized patients most likely to derive survival benefits from its targeted anti-inflammatory effects was essential. The data obtained can guide treatment decisions and resource allocation. The study provides initial evidence assisting clinicians in identifying candidates with severe inflammatory profiles marked by elevated CRP who could achieve optimal outcomes with Tocilizumab administration with limited resources.

6. Confounding Variables: The study should account for potential confounding factors that could influence CRP levels or clinical outcomes, such as comorbidities or other treatments administered alongside Tocilizumab.

As supplementary Table 1 addresses, the main comorbidities were registered and the groups with or without Tocilizumab did not show significant difference.

7. Conflict of Interest: The authors also did not inform us whether there are any conflicts of interest, such as financial ties to the manufacturer of Tocilizumab, which could potentially bias the results or their interpretation.

Thanks for highlighting potential conflicts of interest. We disclosed in the Funding section that no external funding was received for the study, and we declare no conflict of interest. There is no link between the authors and any pharmaceutical laboratory. We acknowledge the importance of explicitly stating no conflicts to emphasize our commitment to unbiased, rigorously ethical knowledge generation.

Finally, I would suggest the authors remove any link connected to their private doc hub/archive such as google drive. All the figures should be provided along with the doc or as a supplementary.

We apologize about the mistake; the link has been removed.

Response for Reviewer 2 comment:

Reviewer #2 (Comments for the Author):

Dear author,

The review of the study titled Tocilizumab Demonstrates Superiority in Decreasing C-Reactive Protein Levels in Hospitalized COVID-19 Patients, compared to Standard Care Treatment Alone has been completed. Your study has some shortcomings due to its retrospective nature, but the part examined has successful results. I have only one suggestion regarding your work.

It is stated in the materials and methods section that patients with bacterial pneumonia were not included in the study. However, in the findings section, it was stated that 65% of the patients received antibiotic treatment. It should be briefly stated why these patients received antibiotics. It will be sufficient to clarify this point.

I wish good work.

Thank you for your essential point regarding antibiotic administration without bacterial pneumonia. Since participants with or without Tocilizumab received a range of immunosuppressive therapies that increase vulnerability to opportunistic bacterial infections, antibiotics were empirically initiated to provide broad coverage when there were signs of potential infection; this was done even if bacterial pneumonia was discarded by initial diagnostic testing but always keeping the balance between risk/benefit use of antibiotics on each patient.

Re: Spectrum02498-23R1 (Tocilizumab Demonstrates Superiority in Decreasing C-Reactive Protein Levels in Hospitalized COVID-19 Patients, compared to Standard Care Treatment Alone,)

Dear Dr. María Cecilia Ishida-Gutierrez:

Thank you for the privilege of reviewing your work. Below you will find my comments, instructions from the Spectrum editorial office, and the reviewer comments.

Revision Guidelines

Sincerely,
Maria Grazia Cusi
Editor
Microbiology Spectrum

Reviewer #1 (Comments for the Author):

None

Reviewer #2 (Comments for the Author):

Dear author,

The review of the study titled Tocilizumab Demonstrates Superiority in Decreasing C-Reactive Protein Levels in Hospitalized COVID-19 Patients, compared to Standard Care Treatment Alone, after corrections, has been completed. Some minor points need to be reconsidered. These points are:

1. It would be more appropriate to write the keywords in alphabetical order.
2. "In other studies, the severe cases were defined as patients with <92% of oxygen saturation at room air and CRP {greater than or equal to}75 mg/L (4), or the admission to the intensive care unit (ICU; 5)."

In the sentence above, the expression "ICU" and the source number should be placed in different parentheses.

3. In the first sentence of the discussion section, "COVID-19 patients." The statement should be removed.

I wish good work.

Dear Reviewers:

Thank you for your comments for our manuscript, please find below the response to them.

Reviewer #1 (Comments for the Author):

None

Reviewer #2 (Comments for the Author):

Dear author,

The review of the study titled Tocilizumab Demonstrates Superiority in Decreasing C-Reactive Protein Levels in Hospitalized COVID-19 Patients, compared to Standard Care Treatment Alone, after corrections, has been completed. Some minor points need to be reconsidered. These points are:

1. It would be more appropriate to write the keywords in alphabetical order.

R. Some key words were eliminated and the remaining were ordered in alphabetical order as suggested.

2. "In other studies, the severe cases were defined as patients with <92% of oxygen saturation at room air and CRP {greater than or equal to}75 mg/L (4), or the admission to the intensive care unit (ICU; 5)."

In the sentence above, the expression "ICU" and the source number should be placed in different parentheses.

R. The "(ICU)" word was better removed, since it will appear on the paragraph below this sentence.

3. In the first sentence of the discussion section, "COVID-19 patients." The statement should be removed.

The statement was removed.

Thank you for your detailed review and comments.

Cecilia Ishida

Re: Spectrum02498-23R2 (Tocilizumab Demonstrates Superiority in Decreasing C-Reactive Protein Levels in Hospitalized COVID-19 Patients, compared to Standard Care Treatment Alone,)

Dear Dr. María Cecilia Ishida-Gutierrez:

Your manuscript has been accepted, and I am forwarding it to the ASM production staff for publication. Your paper will first be checked to make sure all elements meet the technical requirements. ASM staff will contact you if anything needs to be revised before copyediting and production can begin. Otherwise, you will be notified when your proofs are ready to be viewed.

Sincerely,
Maria Grazia Cusi
Editor
Microbiology Spectrum